# Exceptional longevity of mammalian ovarian and oocyte macromolecules throughout the reproductive lifespan

**Ewa K Bomba-Warczak[1†], Karen M Velez[2†], Luhan T Zhou[2], Christelle Guillermier[3,4], Seby Edassery[1], Matthew L Steinhauser[3,4], Jeffrey N Savas[1\*], Francesca E Duncan[2\*]**

[1]Department of Neurology, Northwestern University Feinberg School of Medicine, Chicago, United States; [2]Department of Obstetrics and Gynecology, Feinberg School of Medicine, Northwestern University, Chicago, United States; [3]Department of Medicine, Aging Institute, University of Pittsburgh School of Medicine, Pittsburgh, United States; [4]Department of Medicine, Division of Genetics, Brigham and Women's Hospital, Boston, United States

**\*For correspondence:**
jeffrey.savas@northwestern.edu (JNS);
f-duncan@northwestern.edu (FED)

[†]These authors contributed equally to this work

**Competing interest:** The authors declare that no competing interests exist.

## eLife assessment

This **important** study highlights cell types preserving long-lived proteins and lays a foundation for identifying exceptionally long-lived proteins in the ovary. **Convincing** evidence describes helpful data about protein turnover and identifies long-lived macromolecules in oocytes and somatic cells during mouse ovarian aging. This work will be of interest to researchers working on aging and reproductive health.

**Abstract** The mechanisms contributing to age-related deterioration of the female reproductive system are complex, however aberrant protein homeostasis is a major contributor. We elucidated exceptionally stable proteins, structures, and macromolecules that persist in mammalian ovaries and gametes across the reproductive lifespan. Ovaries exhibit localized structural and cell-type-specific enrichment of stable macromolecules in both the follicular and extrafollicular environments. Moreover, ovaries and oocytes both harbor a panel of exceptionally long-lived proteins, including cytoskeletal, mitochondrial, and oocyte-derived proteins. The exceptional persistence of these long-lived molecules suggest a critical role in lifelong maintenance and age-dependent deterioration of reproductive tissues.

## Introduction

The female reproductive system is the first to age in the human body with fertility decreasing for women in their mid-thirties and reproductive function ceasing completely at menopause (*Broekmans et al., 2009*). In the ovary, aging is associated with a loss in gamete quantity and quality which contributes to infertility, miscarriages, and birth defects (*Reefhuis and Honein, 2004*; *Mai et al., 2013*; *Hollier et al., 2000*; *Ann Johnson et al., 2012*). Moreover, the age-dependent loss of the ovarian hormone, estrogen, has adverse general health outcomes (*Buyuk et al., 2010*). These sequelae are significant as women globally are delaying childbearing and the gap between menopause and lifespan is widening due to medical interventions (*Sauer, 2015*; *Balasch and Gratacós, 2012*). Although aging is a multifaceted process, loss of proteostasis and dysfunctional protein quality control pathways are hallmarks of reproductive aging (*Duncan et al., 2017*).

Proteostasis relies on tight inter-regulation of protein synthesis, post-translational modifications, folding, and degradation (*Basisty et al., 2018*). While most protein lifetimes in mammals fall within the scale from hours to days (*Fornasiero et al., 2018*; *Fornasiero and Savas, 2023*), a subset of intracellular proteins persists for months in rodents (*Bomba-Warczak et al., 2021*; *Savas et al., 2012*). These long-lived proteins (LLPs) are enriched in tissues harboring long-lived post-mitotic terminally differentiated cells, such as the brain and heart (*Bomba-Warczak et al., 2021*; *Savas et al., 2012*; *Toyama et al., 2013*). Although the extended lifespan of LLPs places them at inherent risk for accumulating damage during aging, many of them provide key structural support for the lifelong maintenance of highly stable protein complexes in cells (*Bomba-Warczak and Savas, 2022*).

The mammalian ovary is comprised of a fixed and nonrenewable pool of long-lived cells or oocytes. In humans, oocytes initiate meiosis during fetal development, and by birth, all oocytes are arrested at prophase of meiosis I (*Hunt and Hassold, 2008*; *Nagaoka et al., 2012*). This cell cycle arrest is maintained until ovulation, which occurs any time between puberty and menopause, and thus can span decades. The oocytes are particularly sensitive to protein metabolism alterations because they contribute the bulk cytoplasm to the embryo following fertilization. Thus, maternal proteins produced during oogenesis are essential to generate high-quality gametes (*Duncan et al., 2017*). The ovarian microenvironment is a critical determinant of gamete quality and has been shown to become fibroinflamed and stiff with age (*Amargant et al., 2020*; *Briley et al., 2016*; *Machlin et al., 2021*). Although a small number of oocyte-specific proteins have been identified as long-lived, including cohesins and several centromere-specific histones, there has not been a discovery-based approach to define the long-lived proteome of the ovary and oocyte. Thus, the potential contribution of LLPs to the age-related deterioration of the reproductive system in mammals remains to be elucidated. In this study we used multi-generational whole animal metabolic stable isotope labeling and leading mass spectrometry (MS)-based quantitative proteomic approaches to visualize and identify ovarian and oocyte long-lived macromolecules in vivo during milestones relevant to the reproductive system.

## Results

### Exceptional longevity of ovarian structures and molecules in mammals

The mammalian ovary is a structurally complex, heterogenous, and dynamic organ with follicles at different stages of development, remnants of ovulation (corpora lutea), and a heterogeneous stroma (*Kinnear et al., 2020*; *Figure 1A*). Very little is known about the long-term homeostasis and relative turnover of the ovarian tissue components during aging. To address this, we visualized the lifespan of ovarian macromolecules in mammals using a combination of stable isotope labeling and multi-isotope imaging mass spectrometry (MIMS) (*Figure 1—figure supplement 1A*). First, using a two-generational metabolic labeling of animals with $^{15}N$, we generated a cohort of fully $^{15}N$-labeled pups. After birth, labeled females were kept with the labeled dam until weaning, at which time their food source was switched to the $^{14}N$-chow chase. Using previously established methods (*Duncan et al., 2017*; *Kimler et al., 2018*; *Perrone et al., 2023*; *Quan et al., 2020*), we determined that chase periods of 6 and 10 months represented a biologically relevant reproductive aging continuum as mice within this age range begin to manifest ovarian aging phenotypes, including follicle loss (*Figure 1—figure supplement 1B and D*), decreased ovulation (*Figure 1—figure supplement 1E and F*), and increased fibrotic foci in the ovarian stroma (*Figure 1—figure supplement 1G–J*). Importantly, sufficient numbers of oocytes can still be collected at these timepoints for meaningful further downstream analyses of this rare cell type. We first performed MIMS on ovarian sections to visualize and quantify the abundance of $^{14}N$, representing molecules which have been replaced during the chase period (blue-teal), and $^{15}N$, which represents $^{15}N$-containing molecules that must have persisted through the chase period and therefore are long-lived (orange-pink) (*Figure 1B*, *Figure 1—figure supplement 2A and B*; *Steinhauser and Lechene, 2013*; *Guillermier et al., 2014*; *Nuñez et al., 2018*). Within the ovarian follicles, MIMS revealed a strikingly higher abundance of $^{15}N$ containing molecules in primordial and primary stages relative to later stage follicles, suggesting that primordial follicles can persist for months with limited macromolecular turnover (*Figure 1C*). As follicles progress through the primary, secondary, and antral stages the $^{15}N/^{14}N$ ratio decreases due to signal dilution associated with the increase of cell number and follicle growth (*Figure 1C*, *Figure 1—figure supplement 2C*). This change in $^{15}N$ abundance was most apparent in the granulosa cells where long-lived molecules

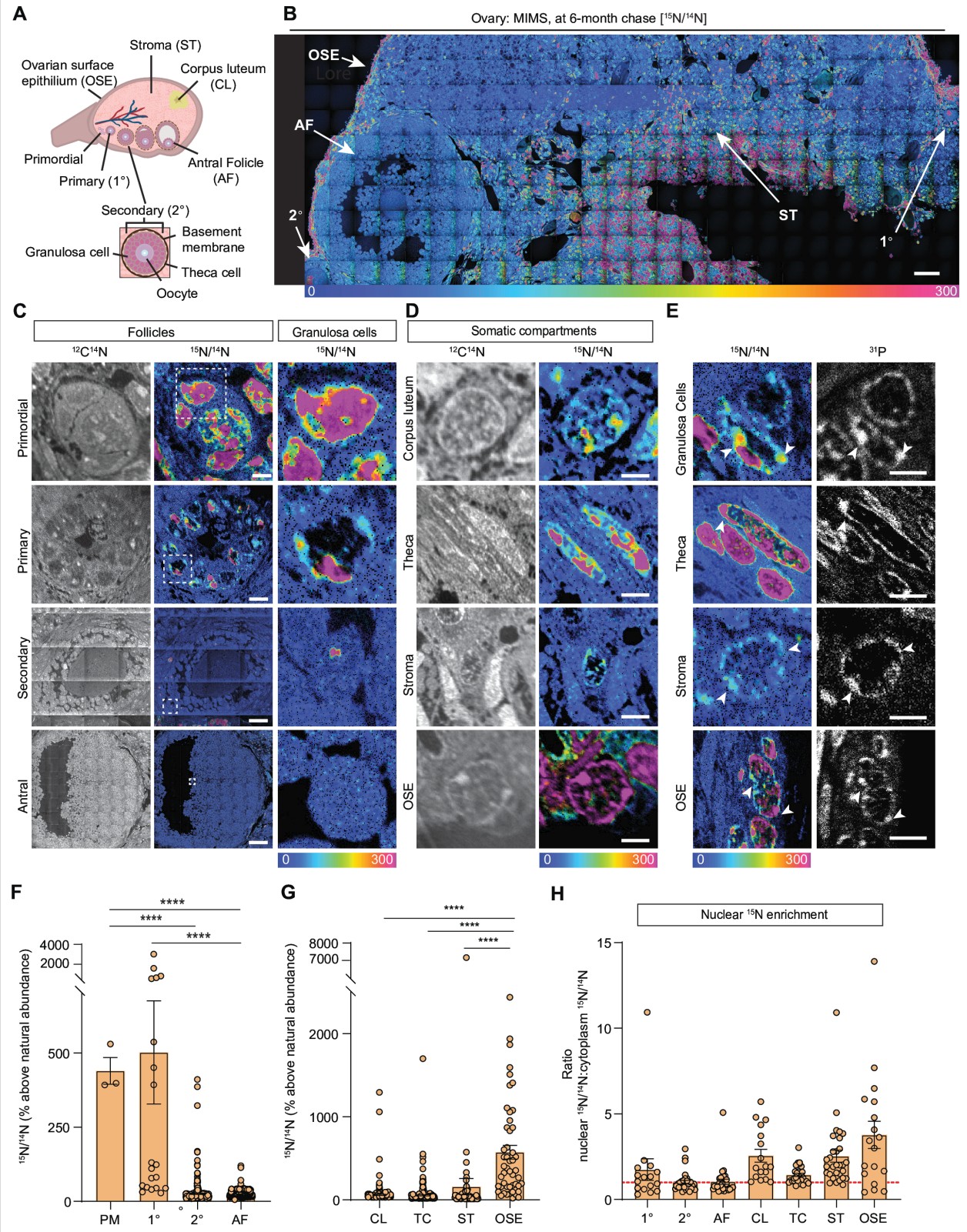

**Figure 1.** Multi-isotope imaging mass spectrometry (MIMS) analysis of cells and structures across ovarian tissue sections. (**A**) Diagram depicting and defining structures of the mammalian ovary. (**B**) Representative hue saturation intensity (HSI) mosaic from an ovary of a $^{15}$N-labeled mouse (6 months of age). Localization and abundance of $^{15}$N varied depending on cell type. HSI scale was set to 0% (natural $^{15}$N/$^{14}$N ratio) to 300% (above the natural ratio). (**C**) High abundance of $^{15}$N is seen in early-stage follicles, specifically within granulosa cells. (**D**) Representative images of somatic cells show differences

*Figure 1 continued on next page*

*Figure 1 continued*

in $^{15}$N-labeling. (**E**) Intracellular abundance of $^{15}$N is colocalized with $^{31}$P abundance across all cell types. (**F**) Differences in $^{15}$N/$^{14}$N ratios reveal granulosa cells of early-stage follicles have greater $^{15}$N abundance than later stages (n=3 (PM), 20 (1°), 305 (2°), 2848 (AF)). (**G**) Among somatic cells, quantitative analysis shows a greater abundance of $^{15}$N at the ovarian surface epithelium (n=242 (CL), 276 (TC), 74 (ST), 74 (OSE)). (**H**) Ratio analysis shows abundance of $^{15}$N localized in nuclear regions of cells (n=16 (1°), 30 (2°), 30 (AF), 17 (CL), 30 (TC), 30 (ST), 18 (OSE)). A hypothetical ratio of one, denoted as a red dash line, signifies no difference in $^{15}$N abundance between cytoplasmic and nuclear regions. Abbreviations: PM (primordial follicle), 1° (primary follicle), 2° (secondary follicle), AF (antral follicle), CL (corpus luteum), TC (theca cell), ST (stroma), and OSE (ovarian surface epithelium). HSI scale for all images was set to 0%-300% (above natural abundance). Data are shown as mean± SEM. Statistical analysis was performed using a one-way ANOVA. Asterisk denotes statistical significance (* p≤0.05; ** p≤0.01; *** p≤0.001; **** p≤0.0001). Scale bar = (B): 50 µm; (C): 3 µm (PM), 6 µm (1°), 30 µm (2°), 75 µm (AF); (D): 2.5 µm (CL), 2.5 µm (TC), 5 µm (ST), 2.5 µm (OSE); (E): 3 µm (GC), 3 µm (TC), 2.5 µm (ST), 6 µm (OSE).

The online version of this article includes the following figure supplement(s) for figure 1:

**Figure supplement 1.** Multi-generational whole animal pulse-chase labeling design along the reproductive aging continuum.

**Figure supplement 2.** Multi-isotope imaging mass spectrometry (MIMS) uncovers structures enriched with $^{15}$N.

were significantly higher in those within primordial and primary follicles compared to later follicle stages (*Figure 1C and F*). In addition to granulosa cells, long-lived molecules also localized to the basement membrane of some early growing follicles (*Figure 1—figure supplement 1D*). Beyond the follicle, other somatic compartments of the ovary were additionally found to have higher $^{15}$N:$^{14}$N ratio suggesting enrichment in long-lived components, including steroidogenic cells (theca layer and corpora lutea), stromal cells, and cells within the ovarian surface epithelium (OSE) (*Figure 1B and D*). Our quantitative analysis revealed that the OSE had significantly higher $^{15}$N/$^{14}$N ratio among the mentioned cell types (*Figure 1G*). Lastly, we analyzed the $^{15}$N/$^{14}$N ratios in the nucleus relative to the cytoplasm, which revealed significant enrichment of long-lived, $^{15}$N- positive molecules in the nuclei of granulosa cells within primary follicles, and cells within the corpora lutea, the stroma, and the OSE (*Figure 1E and H*). As both proteins and nucleic acids contain nitrogen, the $^{15}$N-positive nuclear signal could correspond to known long-lived nuclear proteins, such as histones, nuclear pore proteins, and lamins, which were previously identified in neuronal, post-mitotic cells (*Savas et al., 2012*; *Toyama et al., 2013*). Alternatively, as these nuclear $^{15}$N-hotspots coincided with the $^{31}$P signal, which is enriched in DNA and correlates with DNA labeling (*Guillermier et al., 2017a*), this data may suggest that the DNA itself is long-lived (*Figure 1E*). These results reveal that distinct macromolecular components within select ovarian cells and tissue regions persist throughout the healthy reproductive stage, with limited renewal, and those long-lived molecules persist through the stage where ovaries manifest marked reproductive aging phenotypes.

## Identification of the long-lived proteome in mammalian ovaries

Although MIMS analysis provides important spatial information on long-lived structures in the ovary, it does not provide the identity of $^{15}$N-containing macromolecules that comprise them. To address this, we performed liquid chromatography mass spectrometry (LC-MS/MS)-based proteomic analysis of ovarian tissues isolated from metabolically labeled mice. After 6 months of $^{14}$N-chase, we identified 36,222±6768 $^{14}$N-peptides mapping to 4106 proteins, and 13±5 $^{15}$N-peptides, which collectively mapped to 33 LLPs across all the biological replicates (*Figure 2A and B*, *Supplementary file 1*). To gain a deeper insight into the persistence of LLPs beyond 6 months, we also analyzed ovaries isolated from females that remained on $^{14}$N chase for 10 months. Although the total numbers of both $^{14}$N and $^{15}$N-peptides and proteins were similar between the two timepoints (39,637±12005 $^{14}$N-peptides mapping to 4464 proteins and 13±6 $^{15}$N-peptides mapping to 15 LLPs), the majority of LLPs identified at 6 months were no longer identified as long-lived at the 10-month chase timepoint. Only tubulins and select histones persisted and were identified as LLPs at this aged timepoint. Gene ontology (GO) enrichment analysis of LLPs identified at the 6-month chase timepoint revealed significant overrepresentation for terms related to chromatin, nucleosome, tubulin complex, and mitochondria (*Figure 2C*).

Next, we determined the quantity of each LLP remaining after the $^{14}$N-chase period by calculating the fractional abundance (FA; $^{15}$N-remaining, $^{15}$N/[$^{14}$N + $^{15}$N]) for each LLP in the ovary using reconstructed MS1 chromatograms from LC-MS/MS analysis (*Park et al., 2008*; *Figure 2D–F*). We found that LLPs had significant differences in FA between the 6- and 10-month timepoints, with 1.13 ± 0.08% and 1.37 ± 0.12% $^{15}$N-remaining, respectively (*Figure 2D*). The discrete pool of proteins that persisted throughout both timepoints allowed for a unique direct comparison of $^{15}$N to $^{14}$N-peptide

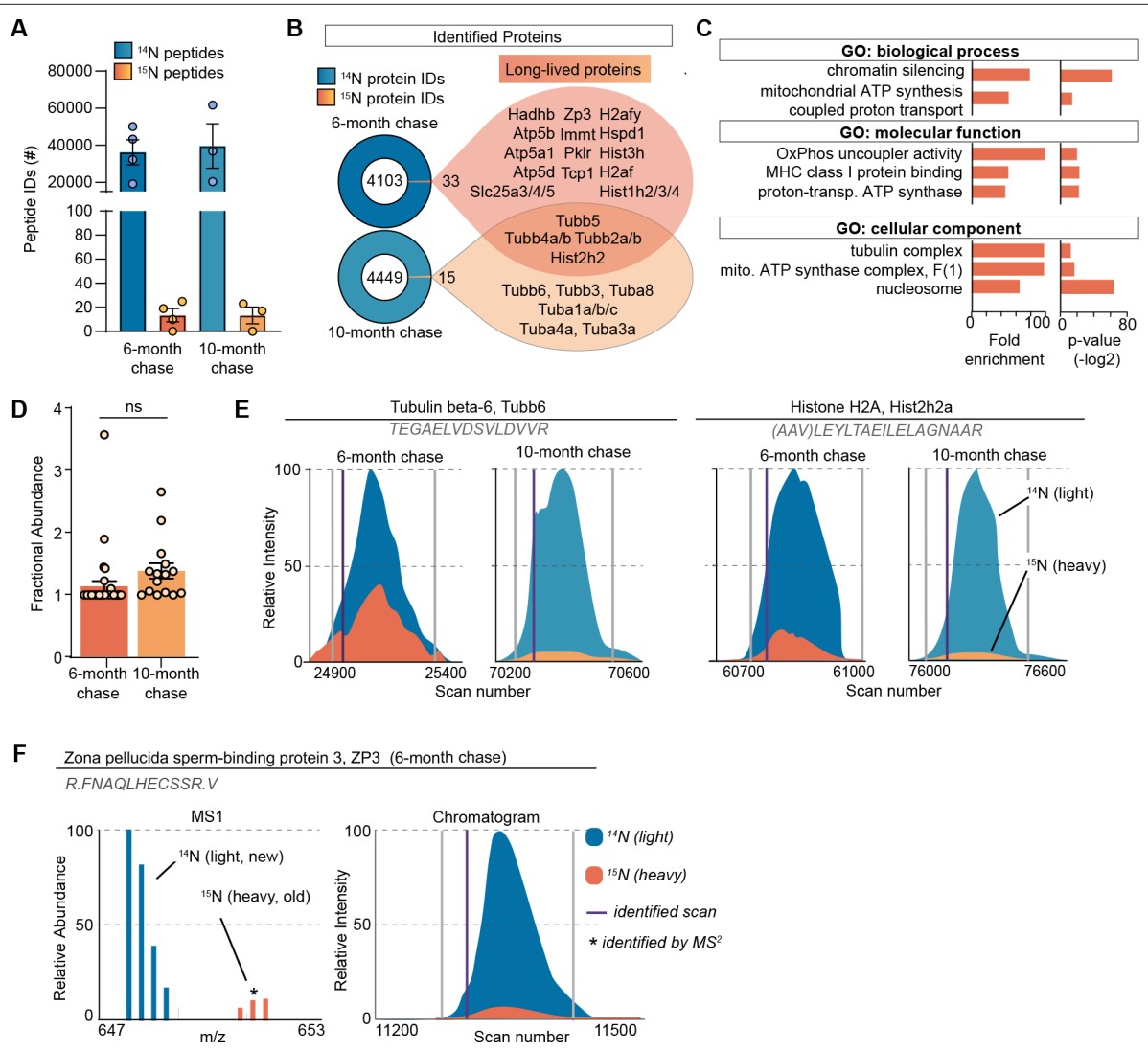

**Figure 2.** Long-lived proteome in mammalian ovaries. (**A**) Summary of peptide identification at 6- and 10-month chase points, blue graphs indicate ¹⁴N-peptide IDs, orange/yellow graphs indicate ¹⁵N-peptide IDs. (**B**) Summary of protein identification at each timepoint, with a list of proteins identified as long-lived in orange. (**C**) Gene ontology (GO) analysis of the long-lived proteins (LLPs) identified in ovaries at 6-month chase revealed that terms related to chromatin, nucleosome, tubulins, and mitochondria are significantly enriched. (**D**) Fractional abundance of LLPs identified at 6- and 10-month chase. (**E**) Annotated representative chromatograms of two representative proteins that persist through both 6- and 10-month chase, illustrating the decreasing ¹⁵N-signal over time. Blue - ¹⁴N (new), orange ¹⁵N (old), purple line: identified scan. (**F**) Annotated representative raw MS1 scan of zona pellucida-3 protein (ZP3). Mean ± SEM; three to four biological replicates per timepoint, ns - not significant by Student's t-test.

peak intensities. This analysis showed reduced abundance of ¹⁵N at the 10-month chase timepoint, consistent with continual, albeit slow, turnover of the protein pool (***Figure 2E***). Interestingly, at the 6-month chase timepoint, we identified ¹⁵N-peptides mapping to an oocyte-specific protein, zona pellucida-3 protein (ZP3), indicating that a pool persists without turnover for at least 6 months, but less than 10 months, as long-lived ZP3 was no longer identified at the 10-month timepoint (***Figure 2F***). While histones and tubulin have been previously identified as LLPs in the mammalian brain and heart tissues, the 6-month long persistence of ZP3 in mouse ovaries is unexpected and of potential importance to reproductive biology.

# Exceptional longevity of mitochondrial and myosin proteins in mammalian oocytes

MIMS analysis of ovarian sections also captured oocytes at various stages of development, which in addition to the enrichment of $^{15}$N-signal within the oocyte nucleus revealed multiple smaller cytoplasmic $^{15}$N-hotspots (*Figure 3A*). Thus, to determine the identity of these $^{15}$N-enriched molecules we isolated fully grown oocytes from ovaries of labeled mice at 6- and 10-month chase timepoints followed by LC-MS/MS analysis (*Figure 3B* and *Figure 3—figure supplement 1A*). In oocytes, we identified a total of 2919 proteins at 6 months and 3234 proteins at 10 months (*Figure 3B*, *Supplementary file 2*). Although after 6 months of chase, we identified 146 LLPs in oocytes, only 11 LLPs were identified after 10 months, indicating that by this timepoint a vast majority of LLPs have been degraded and renewed. Interestingly, the GO analysis of LLPs in oocytes identified at 6-month timepoint revealed a significant enrichment of terms related to nucleosomes, myosin complex, and several additional terms related to mitochondria including OxPhos complexes, mitochondrial nucleoid, TCA cycle complexes, and mitochondrial permeability transition pore complex (*Figure 3B*).

Next, we quantified the fraction of each protein pool that persisted for 6 or 10 months by calculating FA values, where the higher the value the longer-lived the corresponding proteins are (*Figure 3—figure supplement 1B*). Overall, there was no significant difference in FA between the two chase timepoints, with 41±2.9 and 50.8±12.2 $^{15}$N-remaining at 6 and 10 months, respectively (*Figure 3—figure supplement 1C*). The higher average FA average at 10-month chase is likely due to turnover of proteins with lower FA values at 6 months, which by 10 months would leave the oocyte with the most persistent pool of proteins. In agreement, the FA values for four LLPs that were identified at both timepoints (Hba, Atp5a, Atp5B, and Hist1h4a) sharply decline between 6 and 10 months (*Figure 3—figure supplement 1D*), demonstrating protein degradation and replenishment. Considering the high abundance of mitochondrial proteins in our dataset, and recent reports showing that expression of mitochondrial proteins is suppressed in aging oocytes (*Rodríguez-Nuevo et al., 2022*), we compared the number of mitochondrial proteins and spectral counts identified (*Figure 3—figure supplement 1E and F*), as well as FA (*Figure 3—figure supplement 1G*) for mitochondrial proteins across the two timepoints. Our results indicate that while there are no significant differences in total mitochondrial protein abundance at 6- and 10-month post-chase, the pool of long-lived mitochondrial proteins decreased significantly, with the majority of proteins being turned over at the later timepoint.

Hierarchical clustering of the LLPs identified at the 6-month timepoint revealed mitochondrial proteins and myosins as the two protein groups with the highest FA in the oocyte (*Figure 3C*). In particular, mitochondria exhibited a wide range of FA values ranging from 1.10% to 98.9%, with an average of 55.9±35.1 of $^{15}$N-remaining at 6 months (*Supplementary file 2*). FA values for myosins were markedly higher than both actin and tubulin with an average FA value of 80.97 ± 19.8% for myosins, 31.4 ± 15.6% for actins, and only 8.3 ± 1.4% for tubulins. This indicates that while all three cytoskeletal components are long-lived, nearly 81% of the myosin protein pool persists throughout the 6-month timepoint, whereas only 31% of actin protein pool and 8.3% of tubulin protein pool persists throughout the same length of time. Histones were also identified as LLPs in oocytes with average FA values of 7.7±9.8. Mitochondrial proteins, histones, myosins, and tubulins have been previously identified as LLPs in brain and heart tissues, which are known to contain long-lived terminally differentiated cells (i.e. neurons in the brain and cardiac myocytes in the heart) (*Bomba-Warczak et al., 2021*). Importantly, however, this is the first time that a subset of the same proteins has been identified as long-lived in the germ cell (*Figure 3D*). Interestingly, the FA of mitochondrial LLPs in the brain (10.2 ± 6.6%) (*Bomba-Warczak et al., 2021*) and myosins in the heart (4.6 ± 8.1%) (*Bomba-Warczak et al., 2021*) is much lower than in the same LLPs quantified in the oocyte (55.9±35.1) (*Figure 3D*). In contrast, histones were less long-lived in the oocyte compared to the brain, and there was no significant difference observed for tubulins (*Figure 3D*). Thus, although the identity of LLPs may be conserved across tissues with long-lived cells, differences in FA may reflect tissue-specific functions and requirements.

## Discussion

In this study, multi-generational whole animal metabolic stable isotope labeling, paired with multi-modal MS-based quantitative approaches enabled visualization and identification of ovarian and

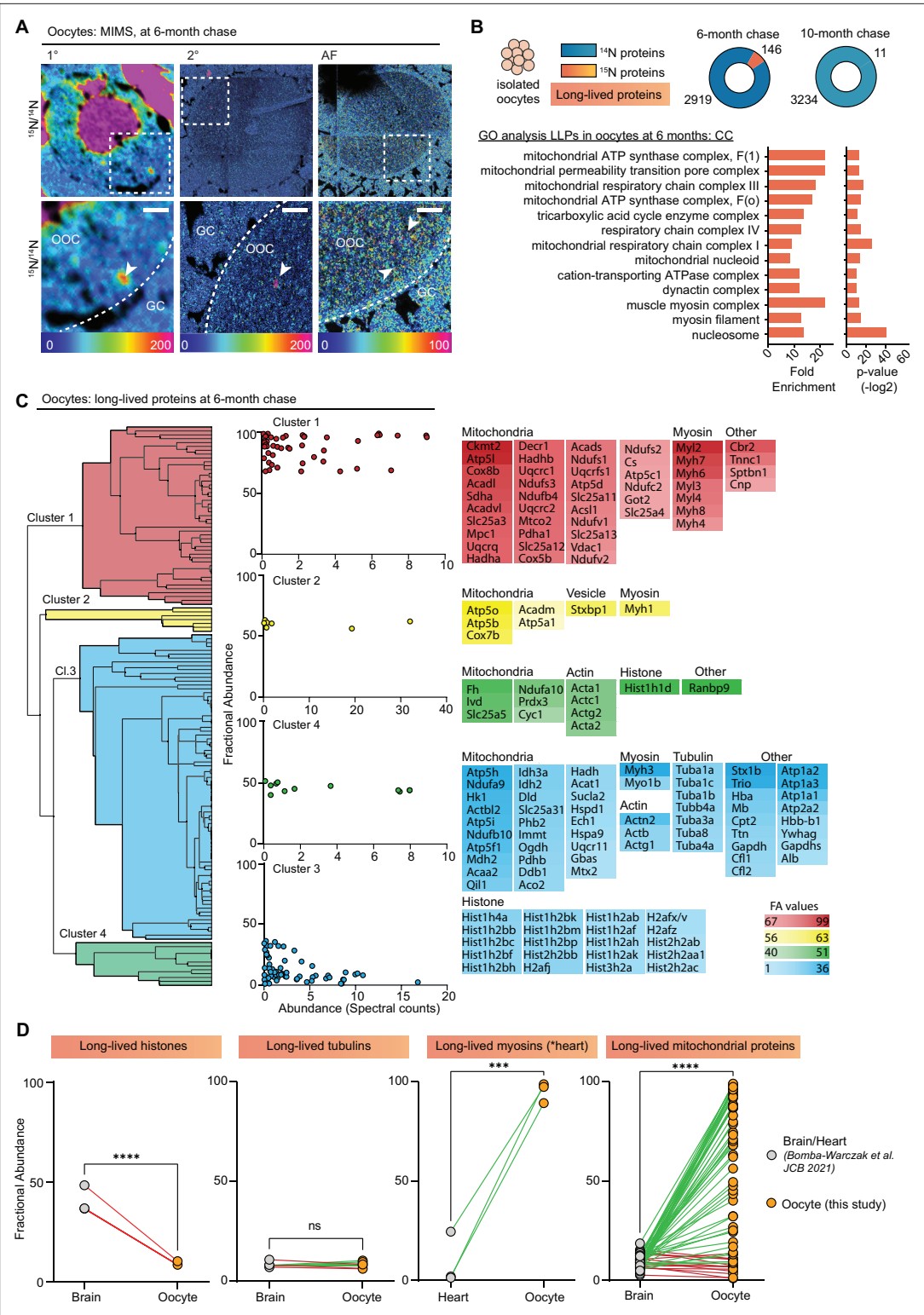

**Figure 3.** Exceptional longevity of nuclear, cytoskeletal, and mitochondrial proteins in mouse oocytes. (**A**) Multi-isotope imaging mass spectrometry (MIMS) analysis reveals high abundance of [15]N in nuclei and throughout the cytoplasm of oocytes. (**B**) Purified oocyte population was harvested from pulse-chased mice and analyzed using gel-liquid chromatography mass spectrometry (LC-MS/MS). Charts illustrate the number of proteins identified at each timepoint (blue) along with long-lived proteins (orange). Gene ontology (GO) analysis of the long-lived proteins (LLPs) identified in oocytes at 6-month chase revealed an enrichment for terms related to nucleosome, myosins, and mitochondria. (**C**) Hierarchical cluster analysis of fractional abundance of LLPs identified in oocytes. (**D**) Direct comparison of fractional abundances of proteins previously identified as long-lived and LLPs in

*Figure 3 continued on next page*

*Figure 3 continued*

oocytes. Mean ± SEM; oocytes collected from four to seven females per timepoint, \*\*\*p-value<0.001, \*\*\*\*p-value<0.0001 by Kruskal-Wallis ANOVA with Tukey's multiple comparisons test. Scale bar = 4 µm.

The online version of this article includes the following source data and figure supplement(s) for figure 3:

**Figure supplement 1.** Long-lived proteins at 6- and 10-month chase points in oocytes.

**Figure supplement 1—source data 1.** PDF containing original scan of oriole-stained gel with annotations.

**Figure supplement 1—source data 2.** PDF containing original scan of oriole-stained gel with annotations.

**Figure supplement 1—source data 3.** Original oriole-stained gel of oocyte homogenate collected at 6 m timepoint.

**Figure supplement 1—source data 4.** Original oriole-stained gel of oocyte homogenate collected at 10 m timepoint.

oocyte long-lived macromolecules in vivo. Our findings provide a novel framework for how long-lived structures may regulate gamete quality. Long-lived macromolecules localized throughout the ovary including the follicular compartment with prominent signals in the granulosa cells of primordial and primary follicles relative to later stage growing follicles. These findings are consistent with the knowledge that the squamous pre-granulosa cells surrounding the oocyte within primordial follicles form early in development which coincided with the $^{15}$N-labeling pulse period. These squamous granulosa cells are generally thought to lack the ability to undergo mitotic division until follicles are activated to grow, so it is not surprising that we observed long-lived macromolecules persisting within them (*Oktay et al., 1997*). In contrast, granulosa cells in growing follicles are generated by cell divisions that take place during follicle activation and growth which coincided with the $^{14}$N-chase period. Thus, long-lived structures (i.e. enriched with $^{15}$N) were diluted through cell divisions during follicle growth. Moreover, during follicle growth, granulosa cells proliferate and differentiate adding new pools of synthesized proteins and molecules. Our results demonstrate that macromolecules formed early in development can persist in squamous granulosa cells for months. Thus, it is possible that these long-lived molecules will accumulate more damage in primordial follicles that remain quiescent for longer periods relative to those that activate earlier. Whether such damage occurs and how it translates into decreased follicle survival or gamete quality will require further investigation.

Within the extrafollicular ovarian environment, the OSE exhibited a striking enrichment of long-lived molecules. The OSE is highly dynamic due to repeated post-ovulation wound healing and repair, and its regenerative capacity occurs through a somatic stem/progenitor cell-mediated process (*Szotek et al., 2008*). Interestingly, LLPs are retained in other cells undergoing repeated asymmetric divisions and are speculated to contribute to the reproductive aging process (*Thayer et al., 2014*). Consistent with this possibility, the architecture and wound healing ability of the OSE is altered with advanced reproductive age (*Mara et al., 2020*). Furthermore, nuclear enrichment of the $^{15}$N signal was highest in cells of the OSE. It is plausible that the older template DNA is segregated into the daughter cell destined to become the stem cell to ensure genetic stability of the OSE. Better understanding of the dynamics of long-lived molecules in the OSE will require generation of specific samples at precise stages of the estrous cycle and across a time course of ovulation to capture follicular rupture and repair.

Through LC-MS/MS analysis, we identified specific LLPs in the mammalian ovary across the reproductive lifespan. LLPs tend to be part of large protein complexes and include histones, nuclear pore complex proteins, lamins, myelin proteins, and mitochondrial proteins (*Bomba-Warczak and Savas, 2022*). In the ovary, the major categories of LLPs included histones, cytoskeletal proteins, and mitochondrial proteins. ZP3 was an oocyte-derived protein identified to be long-lived for at least 6 months. ZP3 is a protein that comprises the zona pellucida (ZP) or glycoprotein matrix of the oocyte, and it is expressed in oocytes of actively growing follicles beginning at the primary stage when the ZP begins to form (*Paulini et al., 2014*). However, during the pulse period, there would have been very few growing follicles in the ovary because of the immature age of the mice, and most importantly, none of these follicles would have persisted 6 months since folliculogenesis only takes approximately 21 days (*Eppig et al., 2002*). These findings suggest that ZP3 may be expressed earlier in oocyte development than previously anticipated. Because LLPs can be at the core of scaffold complexes, a primitive zona may exist at the primordial follicle stage upon which the bona fide ZP is established in growing follicles (*Grootenhuis and Philipsen, 1996*; *Gook et al., 2008*). Consistent with this, expression of ZP proteins has been observed in human primordial follicles (*Gook et al., 2008*). Interestingly, there

are documented age-related defects in the structure and function of the ZP which occur with time-dependent scaffold deterioration. An alternate explanation for our observation is that ZPs from atretic follicles persist and become incorporated into the ovarian matrix. Precedent for this exists because ZP proteins have been identified as components of the matrisome of decellularized porcine ovaries (*Henning et al., 2019*). Interestingly we did not identify ZP3 as an LLP in isolated fully grown oocytes which provides further support that in the ovary, the long-lived pool of ZP3 is derived from primordial follicles or are within the matrix. These possibilities require further investigation and may not be mutually exclusive.

Most LLPs were degraded and replaced between 6 and 10 months of chase. At the 6-month timepoint, we detected more long-lived proteins than the 10-month timepoint in both the ovary and the oocyte because proteins degrade over time, and more time has elapsed at the later timepoint. Moreover, between the 6- and 10-month timepoints, age-related tissue dysfunction is already evident in the ovary. For example, in 6- to 9-month-old mice, there is already a deterioration of chromosome cohesion in the egg which results in increased interkinetochore distances (*Chiang et al., 2010*), and by 10 months, there are multinucleated giant cells present in the ovarian stroma which is consistent with chronic inflammation (*Briley et al., 2016*). Thus, our results suggest that important shifts in the proteome occur during mid to advanced reproductive and may be another early feature of ovarian aging. Whether the LLPs at the 6-month timepoint serve as a protective mechanism in maintaining gamete quality or whether they contribute to decreased quality associated with reproductive aging is an intriguing dichotomy which will require further investigation.

A small subset of tubulins and histones persisted throughout the entire 10-month chase period, indicating that their replacement is exceptionally slow. LLPs which were present at both timepoints and persisted to at least 10 months may have important roles in the aging process. Interestingly, Tubb5 and Tubb4a have high homology to primate-specific Tubb8, and Tubb8 mutations in women are associated with meiosis I arrest in oocytes and infertility (*Dong et al., 2023*; *Feng et al., 2016*). Thus, perturbation of these particular proteins by virtue of their long-lived nature may be associated with impaired function and poor reproductive outcomes, and these possibilities warrant future investigation. We noted that LLPs identified at 10 months were not always identified as long-lived at 6 months. This is a common limitation of MS-based proteomics where each sample is prepared and run individually, which introduces variability between biological replicates, especially with respect to low abundant proteins. To compensate for this known and inherent variability, we applied stringent filtering criteria where we required long-lived peptides to be identified in an independent MS scan which provided us peptides of highest confidence.

By isolating fully grown oocytes from the ovary, we were able to determine the long-lived proteome of a purified germ cell population across the reproductive lifespan. Although we identified certain histone proteins as long-lived, their relative FA was much lower than in the brain, a tissue which also contains post-mitotic cells (*Savas et al., 2012*; *Toyama et al., 2013*). However, histone-variant exchange occurs continuously during mouse oogenesis and is required for both transcriptional regulation and de novo DNA methylation (*Nashun et al., 2015*). Thus, turnover and exchange of histones is likely more dynamic than previously assumed in terminally differentiated or post-mitotic cells, and our findings are consistent with this in the oocyte. Myosin and actin were also identified as LLPs with relatively high FA indicating exceptional longevity beyond 6 months. These proteins play numerous roles in oocyte maturation, fertilization, and egg activation, including nuclear positioning, spindle rotation and anchoring, chromosome segregation, cytokinesis, cortical granule exocytosis, and cytoplasmic flow (*Amargant et al., 2020*; *Uraji et al., 2018*; *Duan and Sun, 2019*; *Roeles and Tsiavaliaris, 2019*; *Ajduk et al., 2011*; *Dunkley et al., 2022*). Defects in many of these processes have been reported with advanced reproductive age, but whether long-lived pools of actin and myosins contribute to this etiology remains to be elucidated (*Díaz and Esponda, 2004*). Interestingly, F-actin stabilization restricts chromosome segregation errors due to cohesion loss which increase with age, so long-lived pools of actin may confer a beneficial effect and protect against aneuploidy (*Dunkley et al., 2022*). The molecular mechanism(s) governing the longevity of the specific identified proteins in oocytes and ovaries as well as their relationship to the age-related decline in fertility and ovarian function require further investigation.

Mitochondrial proteins were the predominant LLPs in isolated oocytes across the reproductive lifespan, and it is plausible that $^{15}$N cytoplasmic hotspots in oocytes that were observed in MIMS may

correspond to mitochondria. Various aspects of mitochondrial dysfunction have been long implicated in the age-dependent decline in gamete quality, with an age-dependent decrease in the number of mitochondria, an increase in abnormal morphology, and altered subcellular distribution (*van der Reest et al., 2021*). Moreover, mtDNA copy number decreases with age, whereas mtDNA mutations increase (*Kujoth et al., 2005*; *Wai et al., 2010*). Finally, the functional capacity of mitochondria decreases with age with decreased membrane potential, increased reactive oxygen species, increased oxidative stress, and decrease energy output (*Babayev and Seli, 2015*). Although it is possible that oocyte long-lived mitochondrial proteins deteriorate with age and contribute to mitochondrial dysfunction, a different model is emerging for these proteins. Mitochondrial proteins are exceptionally long-lived in tissues containing long-lived terminally differentiated cells and have now been documented in the brain (neurons), heart (cardiac myocytes), and the ovary (oocytes) (*Bomba-Warczak et al., 2021*). These mitochondrial LLPs are primarily localized to the cristae invaginations, which are throughout to serve as long-term stable ultrastructure within mitochondria. This strategic enrichment is hypothesized to serve as a lifelong structural pillar of mitochondria to support and maintain these organelles over long time frames (*Bomba-Warczak and Savas, 2022*). Although mitochondrial LLPs persist for at least 6 months in oocyte, the majority were undetectable by 10 months. Thus, it is tempting to speculate that a stable pool of mitochondrial LLPs provides structural support for the maintenance of mitochondrial structure and function early during the reproductive lifespan. However, turnover of mitochondrial LLPs later in the reproductive lifespan may serve as a biological timer of aging. Because mitochondria are maternally inherited, these mitochondrial LLPs formed during fetal development of the mother are likely transferred to the embryo and impact subsequent generations (*Chiaratti et al., 2018*). Further investigation into the role of long-lived mitochondrial proteome in oocytes is necessary to understand their lifelong contribution to reproductive health and fertility outcomes.

## Materials and methods

### Animals

Mice of the FVB strain were obtained from Jackson Laboratory (Bar Harbor, ME, USA). For the whole animal isotope pulse-chase labeling strategy, female (5 weeks of age) and male (10 weeks of age) mice were obtained. To validate the biological relevance of our chase period, reproductive aging parameters were evaluated in unlabeled female mice at 6 weeks, 6 months, and 10 months of age. Retired female FVB mouse breeders were obtained at 15 weeks of age and aged out to 6 and 10 months. Mice were acclimated in the vivarium upon arrival for at least 2 weeks prior to experimental use. All mice were housed at Northwestern University's Center for Comparative Medicine under constant temperature, humidity, and light (14 hr light/10 hr dark). Mice were fed and provided with water ad libitum. All animal care and experimental protocols in this study were conducted under the guidelines set by the NIH Guide for the Care and Use of Laboratory Animals handbook, and the animal protocol was approved by the Animal Care and Use Committee of Northwestern University.

### Pulse-chase labeling strategy

Mice were metabolically labeled using a two-generation metabolic pulse-chase labeling strategy as previously described (*Bomba-Warczak et al., 2021*). Briefly, three female FVB mice were fed a spirulina $^{15}N$-containg chow for the duration of the study (initial labeling, breeding, pregnancy, and weaning) (Cambridge Isotope Laboratories, Inc, Tewksbury, MA, USA). After 13 weeks of being on a $^{15}N$ diet, these labeled females were co-housed with unlabeled males for 5 days to allow a complete estrous cycle for breeding. Pregnant females were housed separately to allow for accurate dating of litter births. All females were allowed to breed a total of three to five times to obtain sufficient pups for the chase period. The pulse period was defined as the timespan between gestation and weaning of pups. Thus, pups conceived from labeled females had nitrogen-containing molecules and proteins labeled with $^{15}N$. At 22 days of age, pups were weaned and fed a $^{14}N$ diet ad libitum. Pups were maintained on a $^{14}N$ diet (chase) until they reached 7 and 11 months of age at which point tissues were harvested for downstream analyses. This animal labeling strategy generated 19 female pups. For ovary studies, four mice were used at 6 months' post-chase and four mice at 10 months' post-chase. One ovary per mouse was used for MIMS, and the contralateral ovary was used for LC-MS/MS. For oocyte studies, four mice were hyperstimulated at 6 months' post-chase and seven mice at 10 months' post-chase.

A total of 104 and 45 oocytes were collected at 6 and 10 months' post-chase, respectively, and these samples were used for LC-MS/MS.

## Ovary and oocyte collection

Ovaries were harvested and placed in a dish containing pre-warmed Leibovitz's medium (L15) (Life Technologies Corporation, Grand Island, NY, USA) supplemented with 3 mg/mL polyvinylpyrrolidone (PVP) (Sigma-Aldrich, St. Louis, MO, USA), 0.5% penicillin-streptomycin (Life Technologies) (L15-PVP). Ovaries were either processed as described below for downstream analyses or used to isolate oocytes. Ovaries used to assess reproductive aging parameters were fixed for histological analysis in Modified Davidson's (Electron Microscopy Sciences, Hatfield, PA, USA) at room temperature (RT) for 2–4 hr with agitation and overnight at 4°C. Ovaries used for MIMS were cut in half and fixed in 2% glutaraldehyde (Electron Microscopy Sciences, Hatfield, PA, USA) and embedded in LR white (EMS). Ovaries used for LC-MS/MS were snap-frozen and stored at –80°C until use.

To maximize the yield of a synchronized population of fully grown oocytes for LC-MS/MS, mice were hyperstimulated with an intraperitoneal injection of 5 IU pregnant mare serum gonadotropin (PMSG) (Prospec Bio, East Brunswick, N). Ovaries were harvested 44–48 hr post-PMSG injection and placed in L15-PVP supplemented with 0.025% milrinone to maintain oocyte meiotic arrest (Sigma-Aldrich, St. Louis, MO, USA).

Cumulus-oocyte-complexes (COCs) were released by puncturing antral follicles with insulin syringes. Oocytes were mechanically denuded from cumulus cells using a 75 µm stripper tip and washed thoroughly in L15/PVP/PS before being snap-frozen. For analysis of ovulated eggs, mice were hyperstimulated as described above with PMSG followed by superovulation induction 44–46 hr later with an intraperitoneal injection of 5 IU human chorionic gonadotropin (hCG) (Sigma, St. Louis, MO, USA). COCs were then collected from the oviducts of each mice 14–16 hr post-hCG injection, and the number of ovulated eggs was counted.

## Ovarian follicle counts and histological analysis

The decline in the number of ovarian follicles is a hallmark of reproductive aging (*Broekmans et al., 2009*). Therefore, we evaluated follicle counts in histological sections of ovaries as done previously (*Duncan et al., 2017*). In brief, fixed ovaries were washed three times with 70% ethanol and processed, dehydrated, and paraffin-embedded using an automated tissue processor (Leica Biosystems, Buffalo Grove, IL, USA). After embedding, ovaries were serial sectioned with every fifth tissue section stained with hematoxylin and eosin (H&E). All H&E-stained tissue sections were digitally scanned at the University of Washington's Histology and Imaging Core using Hamamatsu-HT imaging system (Hamamatsu Photonics, Hamamatsu City, Japan) at ×20 magnification. Scanned images were uploaded and visualized using the NDP.view2 software (Hamamatsu Photonics, Hamamatsu City, Japan). Follicles were classified and counted in every fifth tissue section. Follicles were classified by stage (primordial, primary, secondary, and antral) according to established criteria (*Duncan et al., 2017*). Primordial follicles were comprised of an incomplete layer of squamous granulosa cells, while primary follicles contained a complete layer of cuboidal granulosa cells. Secondary follicles contained two or more layers of cuboidal granulosa cells. Antral follicles had more than eight layers of granulosa cells with the presence of an antrum, or fluid-filled cavity. All primordial and primary follicles were counted regardless of whether the oocyte's nucleus was visible. Secondary and antral follicles were counted only if the nucleus was visible to avoid double counting. Only healthy follicles were included in the final counts. Atretic follicles containing abnormally shaped oocytes with dark, pyknotic granulosa cells were excluded. The average number of follicles per area of ovarian section was calculated and used to compare counts between each age cohort.

Ovarian fibrosis, characterized by excess collagen, is another hallmark of reproductive aging (*Amargant et al., 2020*). We evaluated fibrosis in ovaries by Picrosirius Red (PSR), a histological stain which detects collagen I and III (*Amargant et al., 2020*). Ovarian tissue sections were deparaffinized in Citrisolv (Fisher Scientific, Pittsburgh, PA, USA) and rehydrated in 100%, 70%, and 30% ethanol baths. Slides were submerged in PSR staining solution composed of Sirius Red F3BA (Direct Red 80, C.I. 357.82, Sigma-Aldrich, St. Louis, MO, USA) and picric acid (Sigma-Aldrich, St. Louis, MO, USA) at 0.1% wt/vol for 40 min at RT. The slides were then incubated in acidified water made of 0.05 M hydrochloric acid (Fisher Scientific) for 90 s. Tissue sections were dehydrated in 100% ethanol baths,

three times for 30 s incubations. After dehydration, slides were immersed in Citrisolv for 5 min and mounted with Cytoseal XYL (Fisher Scientific). PSR-stained sections were then imaged with an EVOS FL Auto Imaging system (Thermo Fisher, Waltham, MA, USA) using a ×20 objective. Scans of whole ovarian tissue sections were performed to quantify the area of positive PSR staining using a threshold feature on ImageJ as previously described (*Briley et al., 2016*). PSR-positive staining was analyzed on two different ovarian tissue sections for three mice within each age cohort: 6 weeks, 6 months, and 10 months. Results were averaged to obtain average percent area of collagen.

## MIMS and data processing

Fixed LR white-embedded ovaries were sectioned to 0.5 μm and mounted on silicon wafers. At the Brigham and Women's Hospital Center for NanoImaging, a NanoSims 50L (CAMECA Instruments Inc, Madison, WI, USA) instrument was tuned to simultaneously measure $^{12}C^{14}N^-$, $^{12}C^{15}N^-$, and $^{31}P^-$ secondary ions as described previously for imaging of a wide range of mouse and human tissues (*Steinhauser et al., 2012*). Quantitative $^{12}C^{14}N$ images were used for histological representation of stereotypical ovarian structures and associated cell types. $^{31}P$ images provided additional histological detail and were used for identification of nuclei due to the high phosphorus content of chromatin (*Guillermier et al., 2017a*). Quantitative mass images for $^{12}C^{14}N$ and $^{12}C^{15}N$ were used to generate quantitative $^{15}N{:}^{14}N$ ratio images. For imaging of swathes of the tissue section, images were acquired in chain analysis mode of sequential adjacent fields (dimensions of 45 μm × 45 μm or 50 μm × 50 μm). Sequential tiles were stitched together to make mosaic images for visualizing large sections of the ovary (*Figure 1B*). Some features were then imaged at higher resolution with smaller field sizes. All images were processed and further analyzed using the most recent version of the OpenMIMS 2.0 plugin (https://github.com/BWHCNI/OpenMIMS; *BWHCNI, 2018*) (RRID:SCR_001416, version 2.0.) to ImageJ (*Guillermier et al., 2017b*). $^{15}N$-labeling was visualized by a hue saturation intensity (HSI) transformation of the $^{12}C^{15}N/^{12}C^{14}N$ ratio. The color scale of HSI images is set such that the lower blue bound of the scale is at the $^{15}N$ natural abundance of 0.37% (expressed as 0% above background). The upper magenta bound of the scale is set to reveal labeling differences. The quantitative isotope ratio measurements that form the basis for the images and that are used for statistical analyses are not affected by scaling changes that modify the visual appearance of the images (*Figure 1—figure supplement 2A and B*). A combination of $^{14}N$ and $^{31}P$ images were used to manually select regions of interest within each ovarian section (e.g. individual cells or subcellular structures). Structures and cell types were identified based on morphology and their anatomic location. Cells that were not identifiable or were not visualized due to low ion counts were excluded from the analysis. Typical reasons for difficult identifying cells include sectioning artifacts (cracks, wrinkles) or certain cells that were located at the juncture between adjacent imaging fields, where there is often lower yield of secondary ions. This edge effect is seen to variable degrees in the mosaic images as dark regions at the periphery of an imaging field. All quantitative data for $^{15}N$-labeling was presented as the $^{15}N/^{14}N$ ratio (percentage above natural abundance).

## MS sample preparation: ovaries

Isolated ovaries were homogenized directly in 6 M guanidine hydrochloride solution using bead-based Precellys 24 homogenizer, followed by processing with ProteaseMAX according to the manufacturer's protocol. Samples were reduced with 5 mM Tris(2-carboxyethyl)phosphine (TCEP; vortex 1 hr at RT) alkylated in the dark with 10 mM iodoacetamide (IAA; 20 min), diluted with 50 mM ABC and quenched with 25 mM TCEP. Samples were digested with sequencing grade modified trypsin overnight at 37°C with shaking, spun down (15,000×*g* for 15 min at RT), placed in a new tube, and acidified with TFA to a final concentration of 0.1%. A total of 100 μg of digested and acidified sample was fractionated using High pH Reversed-Phase Peptide Fractionation Kit (Pierce, Cat# 84868). Fractions were step eluted in 300 μL buffer of increasing acetonitrile (ACN) concentrations with decreasing concentration of triethylamine (0.1%) as per the manufacturer's instructions. Samples were dried down with vacuum centrifugation for future MS analysis.

## MS sample preparation: GeLC/MS on oocytes

Isolated oocytes were lysed directly in RIPA buffer, mixed with 6× SDS sample buffer, boiled for 5 min, and separated by SDS-PAGE using 10% Tris-glycine gels (Thermo Scientific, Cat# XV00100PK20).

Gels were stained using Oriole fluorescent gel stain solution, scanned using Bio-Rad Chemidoc XRS system, and cut into sections, chopped into 1 mm × 1 mm cubes, and processed for in-gel digestion. The gel separating oocytes collected at t=6 months was cut into 36 individual pieces, whereas the gel separating oocytes collected at t=10 months was cut into 24 pieces. Gel pieces were incubated in 10 mM TCEP (in 50 mM ABC; 1 hr at 37°C). Liquid was replaced by 50 mM IAA (in 50 mM ABC; 45 min at RT in dark), followed by 50 mM TCEP (in 50 mM ABC; 30 min at RT). Gel pieces were washed with 50 mM ABC (3×) and digested with sequencing grade modified trypsin (1 µg in 50 mM ABC, overnight at 37°C, with shaking). The following day, supernatant was collected into new tube and the gel piece were subjected to three rounds of incubations with 50% ACN and 5% FA solution (30 min at RT, with shaking). Supernatant was collected after each incubation, combined, and dried down with vacuum centrifugation. Samples were re-suspended in 0.5% TFA, desalted with Pierce C18 spin columns (Thermo Scientific, Cat# 89873) per the manufacturer's instructions, and dried down with vacuum centrifugation for future MS analysis.

## MS analysis

Dried samples were re-suspended in 20 µL Buffer A (94.875% $H_2O$ with 5% ACN and 0.125% FA) and 3 µg, as determined by microBCA assay (Thermo Scientific, Cat# 23235) of each fraction or sample were loaded via auto-sampler with either Thermo EASY nLC 100 UPLC or UltiMate 3000 HPLC pump, onto a vented Pepmap 100, 75 µm × 2 cm, nanoViper trap column coupled to a nanoViper analytical column (Thermo Scientific) with stainless steel emitter tip assembled on the Nanospray Flex Ion Source with a spray voltage of 2000 V. A coupled Orbitrap Fusion was used to generate MS data. Buffer A contained 94.785% $H_2O$ with 5% ACN and 0.125% FA, and Buffer B contained 99.875 ACN with 0.125% FA. MS parameters were as follows: Ion transfer tube temp = 300°C, Easy-IC internal mass calibration, default charge state = 2 and cycle time = 3 s. Detector type set to Orbitrap, with 60K resolution, with wide quad isolation, mass range = normal, scan range = 300–1500 m/z, max injection time = 50 ms, AGC target = 200,000, microscans = 1, S-lens RF level = 60, without source fragmentation, and datatype = positive and centroid. MIPS was set as on, included charge states = 2–6 (reject unassigned). Dynamic exclusion enabled with n=1 for 30 s and 45 s exclusion duration at 10 ppm for high and low. Precursor selection decision = most intense, top 20, isolation window = 1.6, scan range = auto normal, first mass = 110, collision energy 30%, CID, detector type = ion trap, OT resolution = 30K, IT scan rate = rapid, max injection time = 75 ms, AGC target = 10,000, Q=0.25, inject ions for all available parallelizable time. For ovary samples the chromatographic run was 4.5 hr per fraction, with the following profile of Buffer B: 2% for 7 min, 2–7% for 1 min, 7–10% for 5 min, 10–25% for 160 min, 25–33% for 40 min, 33–50% for 7 min, 50–95% for 5 min, 95% for 15 min, then back to 2% for the remaining 30 min. For the GeLC/MS oocyte samples, the chromatographic run was 75 min per gel section with the following profile of Buffer B: 2–8% for 6 min, 8–24% for 10 min, 24–36% for 20 min, 36–55% for 10 min, 55–95% for 10 min, 95% for 10 min, then back to 2% for remaining 9 min.

## MS data analysis and quantification

Protein identification/quantification and analysis were performed with Integrated Proteomics Pipeline - IP2 (Integrated Proteomics Applications, Inc, San Diego, CA, USA) using ProLuCID (*Eng et al., 1994*; *Xu et al., 2015*), DTASelect2 (*Cociorva et al., 2006*; *Tabb et al., 2002*), Census, and QuantCompare. Spectral raw files were extracted into MS1, MS2 files using RawConverter 1.0.0.0 (http://fields.scripps.edu/downloads.php). The tandem mass spectra were searched against mouse database (downloaded on March 25, 2014). Searched spectra were matched to sequences using the ProLuCID/SEQUEST algorithm (ProLuCID version 3.1) with 50 ppm peptide mass tolerance for precursor ions and 600 ppm for fragment ions. ProLuCID searches included all fully and half-tryptic peptide candidates that fell within the mass tolerance window and had with unlimited mis-cleavages. Carbamidomethylation (+57.02146 Da) of cysteine was considered as a static modification. Peptide/spectrum matches were assessed in DTASelect2 using the cross-correlation score (XCorr), and normalized difference in cross-correlation scores (DeltaCN). Each protein identified was required to have a minimum of one peptide (-p1) of minimal length of six amino acid residues. False discovery rate (FDR) was set to 1% at the protein level, for all experiments. Peptide probabilities and FDR were calculated based on a target/decoy database containing the reversed sequences of all the proteins appended to the target database (*Peng et al., 2003*). Each dataset was searched twice, once against light ($^{14}$N) and then against

heavy ($^{15}$N) protein databases, as described previously (*Savas et al., 2012*). In the light searches, all of the amino acid residues were considered to contain only $^{14}$N nitrogen, while in the heavy searches, all the amino acid residues were considered to contain only $^{15}$N nitrogen. After the results from ProLuCID were filtered using DTASelect2, the assembled search result file was used to obtain quantitative ratios between $^{14}$N and $^{15}$N using the software Census (*Park et al., 2008*; *MacCoss et al., 2003*).

LLPs were identified as previously described, with modifications (*Savas et al., 2012*). Briefly, in order for a protein to be considered as long-lived, the protein had to be identified in our heavy/light search by at least one long-lived peptide ($^{15}$N-peptide). Peptide ratio measurements were filtered in Census based on a correlation threshold, and only peptides with correlation coefficient above 0.5 were used for further analysis. For singleton analysis, we required the $^{14}$N/$^{15}$N ratio to be greater than 5.0 and the threshold score to be greater than 0.5. Identified peptides were further filtered based on their average peptide enrichment, which we set to 0.9, and peptide profile score, which was set to 0.8. Proteins were only identified as long-lived if they had more than three long-lived peptides that passed the above filtering (except for GeLC/MS experiments where one peptide was required). FA were calculated according to the following formula: $^{14}$N values: $FA = 100 - (100*(1/(1+AR)))$, where FA = fractional abundance and AR = area ratio.

### GO analysis

GO analysis was performed using the Pantherdb (*Mi et al., 2019*). The 'query' is defined as proteins identified as long-lived in the analyzed tissue (based on $^{14}$N-peptide identification), and the reference is defined as all proteins identified in the same tissue analyzed ($^{14}$N- and $^{15}$N-peptide identification).

### Statistical analysis

Statistical analyses were conducted using GraphPad Prism, version 9 (GraphPad Software, Inc). A Student's t-test was performed for comparisons between two groups. For comparisons of more than two groups, a one-way ANOVA was used. A one-sample t-test was used to compare $^{15}$N/$^{14}$N ratios of cytoplasmic and nuclear regions to a hypothetical value of 1. Values>1 signified a cell had greater $^{15}$N-labeling abundance in the nucleus compared to the cytoplasm. Values<1 represented cells with a lower $^{15}$N-labeling abundance in the nucleus compared to the cytoplasm. Values equal to 1 represented a cell with a $^{15}$N-labeling abundance that were equivalent in both nuclear and cytoplasmic regions. Data were considered significant with a p-value<0.05 (*p-value<0.05, **p-value<0.01, ***p-value<0.001). Variability of groups was denoted as standard error of mean (SEM).

### Acknowledgements

We acknowledge Hoi Chang Lee for technical assistance with IVF, Farida Korobova for sample preparation for MIMS, Frank Gyngard for MIMS imaging, and Richard Maas for useful discussions.

## Additional information

### Funding

| Funder | Grant reference number | Author |
| --- | --- | --- |
| Eunice Kennedy Shriver National Institute of Child Health and Human Development | R21HD098498 | Jeffrey N Savas Francesca E Duncan |
| National Institute of Neurological Disorders and Stroke | K99NS126639 | Ewa K Bomba-Warczak |
| National Institute of Neurological Disorders and Stroke | F32 NS106812 | Ewa K Bomba-Warczak |
| National Institute on Aging | R21AG072343 | Jeffrey N Savas |

| Funder | Grant reference number | Author |
| --- | --- | --- |

The funders had no role in study design, data collection and interpretation, or the decision to submit the work for publication.

## Author contributions

Ewa K Bomba-Warczak, Conceptualization, Data curation, Formal analysis, Funding acquisition, Investigation, Visualization, Methodology, Writing – original draft, Writing - review and editing; Karen M Velez, Data curation, Formal analysis, Investigation, Methodology, Writing – original draft; Luhan T Zhou, Data curation, Formal analysis, Methodology, Project administration; Christelle Guillermier, Seby Edassery, Data curation, Formal analysis; Matthew L Steinhauser, Formal analysis, Supervision, Validation, Project administration; Jeffrey N Savas, Francesca E Duncan, Conceptualization, Resources, Data curation, Supervision, Funding acquisition, Investigation, Methodology, Writing – original draft, Project administration, Writing - review and editing

## Author ORCIDs

Ewa K Bomba-Warczak ⓘ https://orcid.org/0000-0002-1744-3516
Karen M Velez ⓘ http://orcid.org/0000-0002-9703-1684
Jeffrey N Savas ⓘ https://orcid.org/0000-0002-8173-5580
Francesca E Duncan ⓘ https://orcid.org/0000-0002-3756-9394

## Ethics

All animal care and experimental protocols in this study were conducted under the guidelines set by the NIH Guide for the Care and Use of Laboratory Animals handbook, and the animal protocol was approved by the Animal Care and Use Committee of Northwestern University (IS00020198).

Reviewer #1 (Public Review): https://doi.org/10.7554/eLife.93172.3.sa1
Reviewer #2 (Public Review): https://doi.org/10.7554/eLife.93172.3.sa2
Reviewer #3 (Public Review): https://doi.org/10.7554/eLife.93172.3.sa3
Author response https://doi.org/10.7554/eLife.93172.3.sa4

# Additional files

## Supplementary files

- Supplementary file 1. Summary of proteins identified in 6- and 10-month chase ovaries.
- Supplementary file 2. Summary of proteins identified in 6- and 10-month chase oocytes.
- MDAR checklist

## Data availability

All data generated or analysed during this study are included in the manuscript and supporting files; Figures 2 and 3 source data are provided in *Supplementary files 1 and 2*. RAW MS data has also been deposited at MassIVE under the accession number MSV000092217.

The following dataset was generated:

| Author(s) | Year | Dataset title | Dataset URL | Database and Identifier |
| --- | --- | --- | --- | --- |
| Savas JN, Bomba-Warczak E | 2023 | Exceptional longevity of ovarian and oocyte macromolecules throughout the reproductive lifespan of mammals | https://massive.ucsd.edu/ProteoSAFe/dataset.jsp?accession=MSV000092217 | MassIVE, MSV000092217 |

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
