## [Editor Report · eLife assessment]

This **important** study highlights cell types preserving long-lived proteins and lays a foundation for identifying exceptionally long-lived proteins in the ovary. **Convincing** evidence describes helpful data about protein turnover and identifies long-lived macromolecules in oocytes and somatic cells during mouse ovarian aging. This work will be of interest to researchers working on aging and reproductive health.

---

## [Referee Report · Reviewer #1 (Public Review)]

Summary:

This manuscript by Bomba-Warczak describes a comprehensive evaluation of long-lived proteins in the ovary using a transgenerational diet-derived 15N-labelling in pulse-chased mice. The transgenerational labeling of proteins (and nucleic acids) with 15N allowed the authors to identify regions enriched in long-lived macromolecules at the 6 and 10-month chase time points. The authors also identified the retained proteins in the ovary and oocyte using MS. Key findings include the relative enrichment in long-lived macromolecules in oocytes, pregranulosa cells, CL, stroma, and surprisingly OSE. Gene ontology analysis of these proteins revealed an enrichment for nucleosome, myosin complex, mitochondria, and other matrix-type protein functions. Interestingly, compared to other post-mitotic tissues where such analyses have been previously performed such as the brain and heart, they find a higher fractional abundance of labeled proteins related to the mitochondria and myosin respectively.

Strengths:

A major strength of the study is the combined spatial analyses of LLPs using histological sections with MS analysis to identify retained proteins.

Another major strength is the use of two chase time points allowing assessment of temporal changes in LLPs associated with aging.

The major claims such as an enrichment of LLPs in pregranulosa cells, GCs of primary follicles, CL, stroma, and OSE are soundly supported by the analyses and the caveat that nucleic acids might differentially contribute to this signal is well presented.

The claims that nucleosomes, myosin complex, and mitochondrial proteins are enriched for LLPs are well supported by GO enrichment analysis and well described within the known body of evidence that these proteins are generally long-lived in other tissues.

Weaknesses:

All weaknesses were addressed in the revised manuscript.

Impact of the work:

This work represents the first study addressing the turnover and retention of long-lived protein in the ovary and will be an invaluable resource for the research community, particularly for those studying ovarian aging. This work also raises important unanswered questions worthy of follow-up including interesting findings regarding the timing of turnover of cell types such as the OSE, organelles such as mitochondria, and ECM proteins such as ZP3 and Tubb family proteins. Most striking are the differences between the two timepoints used (6 and 10 months) which lead the authors to infer trajectories and kinetics of replacement of proteins potentially contributing to ovarian longevity or decline. As such I expect the work will contribute to hypothesis generation and stand to have an important impact on the field.

---

## [Referee Report · Reviewer #2 (Public Review)]

Summary:

The manuscript by Bomba-Warczak et al. applied multi-isotope imaging mass spectrometry (MIMS) analysis to identify the long-lived proteins in mouse ovaries during reproductive aging, and found some proteins related to cytoskeletal and mitochondrial dynamics persisting for 10 months.

Strengths:

The manuscript provides a useful dataset about protein turnover during ovarian aging in mice.

Weaknesses:

The study is pretty descriptive and short of further new findings based on the dataset. In addition, some results such as the numbers of follicles and ovulated oocytes in aged mice are not consistent with the published literature.

Comments on revised version:

The authors did not fully address my previous concerns, especially regarding the verification of the identified proteins, and follow-up functional experiments. In addition, it is still unacceptable for me that the number of ovulated oocytes in mice at 6 months of age is only one third of young mice (10 vs 30; Fig. S1E). The most of published literature show that mice at 12 months of age still have ~10 ovulated oocytes. Moreover, based on the follicle counting method used in the present study (Fig. S1D), there are no antral follicles observed in mice at 6 months and 10 months of age, which is not reasonable.

---

## [Referee Report · Reviewer #3 (Public Review)]

Summary:

In this study Bomba-Warczak et al focused on the reproductive aging, and they presented a map for long-lived proteins which were stable during the reproductive lifespan. The authors used MIMS to examine and show distinct molecules in different cell types in the ovary and tissue regions in 6 months mice, and they also used proteomic analysis to present different LLPs in ovaries between these two timepoints in 6 months and 10 months mice; besides, the authors also examined the LLPs in oocytes in 6 months mice and indicated that these were nuclear, cytoskeleton and mitochondria proteins.

Strengths:

Overall, this study provided important information about the pattern of long-lived proteins during aging, which will contribute to the understanding of the defects caused by reproductive aging.

Weaknesses:

12 months mice were not examined as the typical aged model.

Comments on revised version:

The authors responded to my comments and suggestions. Due to the limitation of the manuscript type, most suggestions of my comments in first round could be considered for future studies by the authors.

---

## [Author Response]

The following is the authors’ response to the current reviews.

**Reviewer 2:**
In addition, it is still unacceptable for me that the number of ovulated oocytes in mice at 6 months of age is only one third of young mice (10 vs 30; Fig. S1E). The most of published literature show that mice at 12 months of age still have ~10 ovulated oocytes.

We disagree with the reviewer’s comment, and the concerns raised were not shared by the other reviewers. We have reported our data with full transparency (each data point is plotted). In the current study, we observed an intermediate phenotype in gamete number (assessed by both ovarian follicle counts and ovulated eggs) when comparing 6 month old mice to 6 week or 10 month old mice; this is as expected. It is well accepted that follicle counts are highly mouse strain dependent. Although the reviewer mentions that mice at 12 months have ~10 ovulated oocytes, no actual references are provided nor are the mouse strain or other relevant experimental details mentioned. Therefore, we do not know how these quoted metrics relate to the female FVB mice used in our current study. As clearly explained and justified in our manuscript, we used mice at 6 months and 10 months to represent a physiologic aging continuum.

Moreover, based on the follicle counting method used in the present study (Fig. S1D), there are no antral follicles observed in mice at 6 months and 10 months of age, which is not reasonable.

This statement is incorrect. Antral follicles were present at 6 and 10 months of age, but due to the scale of the y-axis and the normalization of follicle number/area in Fig. S1D, the values are small. The absolute number of antral follicles per ovary (counted in every 5th section) was 31.3 ± 3.8 follicles for 6-week old mice, 9.3 ± 2.3 follicles for 6-month old mice, and 5.3 ± 1.8 follicles for 10-month old mice. Moreover, it is important to note that these ovaries were not collected in a specific stage of the estrous cycle, so the number of antral follicles may not be maximal. In addition, as described in the Materials and Methods, antral follicles were only counted when the oocyte nucleus was present in a section to avoid double counting. Therefore, this approach (which was applied consistently across samples) could potentially underestimate the total number.

The following is the authors’ response to the original reviews.

**Public Reviews:**

**Reviewer #1 (Public Review):**
Summary:This manuscript by Bomba-Warczak describes a comprehensive evaluation of long-lived proteins in the ovary using transgenerational radioactive labelled 15N pulse-chase in mice. The transgenerational labeling of proteins (and nucleic acids) with 15N allowed the authors to identify regions enriched in long-lived macromolecules at the 6 and 10-month chase time points. The authors also identify the retained proteins in the ovary and oocyte using MS. Key findings include the relative enrichment in long-lived macromolecules in oocytes, pregranulosa cells, CL, stroma, and surprisingly OSE. Gene ontology analysis of these proteins revealed enrichment for nucleosome, myosin complex, mitochondria, and other matrix-type protein functions. Interestingly, compared to other post-mitotic tissues where such analyses have been previously performed such as the brain and heart, they find a higher fractional abundance of labeled proteins related to the mitochondria and myosin respectively.

Response: We thank the reviewer for this thoughtful summary of our work. We want to clarify that our pulse-chase strategy relied on a two-generation stable isotope-based metabolic labelling of mice using 15N from spirulina algae (for reference, please see (Fornasiero & Savas, 2023; Hark & Savas, 2021; Savas et al., 2012; Toyama et al., 2013)). We did not utilize any radioactive isotopes.

Strengths:A major strength of the study is the combined spatial analyses of LLPs using histological sections with MS analysis to identify retained proteins.Another major strength is the use of two chase time points allowing assessment of temporal changes in LLPs associated with aging.The major claims such as an enrichment of LLPs in pregranulosa cells, GCs of primary follicles, CL, stroma, and OSE are soundly supported by the analyses, and the caveat that nucleic acids might differentially contribute to this signal is well presented.The claims that nucleosomes, myosin complex, and mitochondrial proteins are enriched for LLPs are well supported by GO enrichment analysis and well described within the known body of evidence that these proteins are generally long-lived in other tissues.Weaknesses:Comment 1: One small potential weakness is the lack of a mechanistic explanation of if/why turnover may be accelerating at the 6-10 month interval compared to 1-6.

Response 1: At the 6-month time point, we detected more long lived proteins than the 10 month time point in both the ovary and the oocyte. We anticipated this because proteins are degraded over time, and substantially more time has elapsed at the later time point. Moreover, at the 6–10-month time point, age-related tissue dysfunction is already evident in the ovary. For example, in 6-9 month old mice, there is already a deterioration of chromosome cohesion in the egg which results in increased interkinetochore distances (Chiang et al., 2010), and by 10 months, there are multinucleated giant cells present in the ovarian stroma which is consistent with chronic inflammation (Briley et al., 2016). Thus, the observed changes in protein dynamics may be another early feature of aging progression in the ovary.

Comment 2: A mild weakness is the open-ended explanation of OSE label retention. This is a very interesting finding, and the claims in the paper are nuanced and perfectly reflect the current understanding of OSE repair. However, if the sections are available and one could look at the spatial distribution of OSE signal across the ovarian surface it would interesting to note if label retention varied by regions such as the CLs or hilum where more/less OSE division may be expected.

Response 2: We agree that the enrichment of long-lived molecules in the OSE is interesting. To make interpretable conclusions about the dynamics of long-lived molecules in the OSE, we would need to generate a series of samples at precise stages of the estrous cycle or ideally across a timecourse of ovulation to capture follicular rupture and repair. These samples do not currently exist and are beyond the scope of this study. However, this idea is an important future direction and it has been added to the discussion (lines 221-223). Furthermore, from a practical standpoint, MIMS imaging is resource and time intensive. Thus, we are not able to readily image entire ovarian sections. Instead, we focused on structures within the ovary and took select images of follicles, stroma, and OSE. We, therefore, do not have a comprehensive series of images of the OSE from the entire ovarian section for each mouse analyzed.

**Reviewer #2 (Public Review):**
Summary:The manuscript by Bomba-Warczak et al. applied multi-isotope imaging mass spectrometry (MIMS) analysis to identify the long-lived proteins in mouse ovaries during reproductive aging, and found some proteins related to cytoskeletal and mitochondrial dynamics persisting for 10 months.

Response: We thank the reviewer for their summary and feedback.

Strengths:The manuscript provides a useful dataset about protein turnover during ovarian aging in mice.Weaknesses:Comment 1: The study is pretty descriptive and short of further new findings based on the dataset. In addition, some results such as the numbers of follicles and ovulated oocytes in aged mice are not consistent with the published literature, and the method for follicle counting is not accurate. The conclusions are not fully supported by the presented evidence.

Response 1: We agree with the reviewer that this study is descriptive. Our goal, as stated, was to use a discovery-based approach to define the long-lived proteome of the ovary and oocyte across a reproductive aging continuum. As the prominent aging researcher, Dr. James Kirkland, stated: “although ‘descriptive’ is sometimes used as a pejorative term…descriptive or discovery research leading to hypothesis generation has become highly sophisticated and of great relevance to the aging field (Kirkland, 2013).” We respectfully disagree with the reviewer that our study is short of new findings. In fact, this is the first time that a stable two-generation stable isotope-based metabolic labelling of mice in combination with two different state-of-the-art mass spectrometry methods has been used to identify and localize long lived molecules in the ovary and oocyte along this particular reproductive aging continuum in an unbiased manner. We have identified proteins groups that were previously not known to be long lived in the ovary and oocyte. Our hope is that this long-lived proteome will become an important hypothesis-generating resource for the field of reproductive aging.

The age-dependent decline in number of follicles and eggs ovulated in mice has been well established by our group as well as others (Duncan et al., 2017; Mara et al., 2020). Thus, we are unclear about the reviewer’s comments that our results are not consistent with the published literature. The absolute numbers of follicles and eggs ovulated as well as the rate of decline with age are highly strain dependent. Moreover, mice can have a very small ovarian reserve and still maintain fertility (Kerr et al., 2012). In our study, we saw a consistent age-dependent decrease in the ovarian reserve (Figure 1 – figure supplement 1 D), the number of oocytes collected from large antral follicles following hyperstimulation with PMSG (used for LC-MS/MS), and the number of eggs collected from the oviduct following hyperstimulation and superovulation with PMSG and hCG (Figure 1 – figure supplement 1 E and F). In all cases, the decline was greater in 10 month old compared to 6 month old mice demonstrating a relative reproductive aging continuum even at these time points.

Our research team has significant expertise in follicle classification and counting as evidenced by our publication record (Duncan et al., 2017; Kimler et al., 2018; Perrone et al., 2023; Quan et al., 2020). We used our established methods which we have further clarified in the manuscript text (lines 395-397). Follicle counts were performed on every 5th tissue section of serial sectioned ovaries, and 1 ovary from 3 mice per timepoint were counted. Therefore, follicle counts were performed on an average of 48-62 total sections per ovary. The number of follicles was then normalized per total area (mm2) of the tissue section, and the counts were averaged. Figure 1 – figure supplement 1 C and D represents data averaged from all ovarian sections counted per mouse. It is important to note that the same criteria were applied consistently to all ovaries across the study, and thus regardless of the technique used, the relative number of follicles or oocytes across ages can be compared.

**Reviewer #3 (Public Review):**
Summary:In this study, Bomba-Warczak et al focused on reproductive aging, and they presented a map for long-lived proteins that were stable during reproductive lifespan. The authors used MIMS to examine and show distinct molecules in different cell types in the ovary and tissue regions in a 6 month mice group, and they also used proteomic analysis to present different LLPs in ovaries between these two timepoints in 6-month and 10-month mice. The authors also examined the LLPs in oocytes in the 6-months mice group and indicated that these were nuclear, cytoskeleton, and mitochondria proteins.

Response: We thank the reviewer for their summary and feedback.

Strengths:Overall, this study provided basic information or a 'map' of the pattern of long-lived proteins during aging, which will contribute to the understanding of the defects caused by reproductive aging.Weaknesses:Comment 1: The 6-month mice were used as an aged model; no validation experiments were performed with proteomics analysis only.

Response 1: We did not select the 6-month time point to be representative of the “aged model” but rather one of two timepoints on the reproductive aging continuum – 6 and 10 months. In the manuscript (Figure 1 – figure supplement 1) we have demonstrated the relevance of the two timepoints by illustrating a decrease in follicle counts, number of fully grown oocytes collected, and number of eggs ovulated as well as a tendency towards increased stromal fibrosis (highlighted in the main text lines 78-85). Inclusion of the 6-month timepoint ultimately turned out to be informative and essential as many long-lived proteins were absent by the 10 month timepoint. These results suggest that important shifts in the proteome occur during mid to advanced reproductive age. The relevance of these timepoints is mentioned in the discussion (lines 247-270).

Two independent mass spectrometry approaches (MIMS and LC-MS/MS) were used to validate the presence of long-lived macromolecules in the ovary and oocyte. Studies focused on the role of specific long-lived proteins in oocyte and ovarian biology as well as how they change with age in terms of function, turnover, and modification are beyond the scope of the current study but are ongoing. We have acknowledged these important next steps in the manuscript text (lines 286-288, 311-312).

It is important to note, that oocytes are biomass limited cells, and their numbers decrease with age. Thus, we had to select ages where we could still collect enough from the mice available to perform LC-MS/MS.

Recommendations for the authors:Reviewer #1 (Recommendations For The Authors):Comment 1: The writing and figures are beautiful - it would be hard to improve this manuscript.

Response 1: We greatly appreciate this enthusiastic evaluation of our work.

Comment 2: In Fig S1E/F it would help to list the N number here. Why are there 2 groups at 6-12 wk?

Response 2: We did not have 6 month and 10-month-old mice available at the same time to be able to run the hyperstimulation and superovulation experiment in parallel. Therefore, we performed independent experiments comparing the number of eggs collected from either 6-month-old or 10 month old mice relative to 6-12 week old controls. In each trial, eggs were collected from pooled oviducts from between 3-4 mice per age group, and the average total number of eggs per mouse was reported. Each point on the graph corresponds to the data from an individual trial, and two trials were performed. This has been clarified in the figure legend (lines 395-397). Of note, while addressing this reviewer’s comments, we noticed that we were missing Materials and Methods regarding the collection of eggs from the oviduct following hyperstimulation and superovulation with PMSG and hCG. This information has now been added in Methods Section, lines 477-481.

Comment 3: The manuscript would benefit from an explanation of why the pups were kept on a 1-month N15 diet after birth, since the oocytes are already labeled before birth, and granulosa at most by day 3-4. Would ZP3 have not been identified otherwise?

Response 3: The pups used in this study were obtained from fully labeled female dams that were maintained on an15N diet. These pups had to be kept with their mothers through weaning. To limit the pulse period only through birth, the pups would have had to be transferred to unlabeled foster mothers. However, this would have risked pup loss which would have significantly impacted our ability to conduct the studies given that we only had 19 labeled female pups from three breeding pairs. We have clarified this in the manuscript text in lines 78-80. It is hard to know, without doing the experiment, whether we would have detected ZP3 if we only labeled through birth. The expression of ZP3 in primordial follicles, albeit in human, would suggest that this protein is expressed quite early in development.

Comment 4: What is happening to the mitochondria at 6-10 months? Does their number change in the oocyte? Is there a change in the rate of fission? Any chance to take a stab at it with these or other age-matched slides?

Response 4: The reviewer raises an excellent point. As mentioned previously in the Discussion (lines 290-301), there are well documented changes in mitochondrial structure and function in the oocyte in mice of advanced reproductive age. However, there is a paucity of data on the changes that may happen at earlier mid-reproductive age time points. From the oocyte mitochondrial proteome perspective, our data demonstrate a prominent decline in the persistence of long-lived proteins between 6 and 10 months, and this occurs in the absence of a change in the total pool of mitochondrial proteins (both long and short lived populations) as assessed by spectral counts or protein IDs (figure below). These data, which we have added into Figure 3 – figure supplement 1 and in the manuscript text (lines 164-170) are suggestive of similar numbers of mitochondria at these two timepoints. It would be informative to do a detailed characterization of oocyte mitochondrial structure and function within this window to see if there is a correlation with this shift in long lived mitochondrial proteins. Although this analysis is beyond the scope of the current manuscript, it is an important next line of inquiry which we have highlighted in the manuscript text (lines 255-257 and 311-312).

**Reviewer #2 (Recommendations For The Authors):**
Several concerns are raised as shown below.Comment 1: In Fig. 2F, it is surprising that ZP3 disappeared in the ovary from mice at the age of 10 months by MIMS analysis, because quite a few oocytes with intact zona pellucida can still be obtained from mice at this age. Notably, ZP would not be renewed once formed.

Response 1: To clarify, Figure 2F shows LC-MS/MS data and not MIMS data. As mentioned in the Discussion, the detection of long-lived pools of ZP3 at 6 months cannot be derived from newly synthesized zona pellucidae in growing follicles because they would not have been present during the pulse period. The only way we could detect ZP3 at 6 months is if it forms a primitive zona scaffold in the primordial follicle or if ZPs from atretic follicles of the first couple of waves of folliculogenesis incorporate into the extracellular matrix of the ovary. The lack of persistence of ZP3 at 10 months could be due to protein degradation. Should ZP3 indeed form a primitive zona, its loss at 10 months would be predicted to result in poor formation of a bona fide zona pellucida upon follicle growth. Interestingly, aging has been associated with alterations in zona pellucida structure and function. These data open novel hypotheses regarding the zona pellucida (e.g. a primitive zona scaffold and part of the extracellular matrix) and will require significant further investigation to test. These points are highlighted in the Discussion lines 227-245.

Comment 2: To determine whether those proteins that can not be identified by MIMS at the time point of 10 months are degraded or renewed, the authors should randomly select some of them to examine their protein expression levels in the ovary by immunoblotting analysis.

Response 2: To clarify, proteins were identified by LC-MS/MS and not MIMS which was used to visualize long lived macromolecules. Each protein will be comprised of old pools (15N containing) and newly synthesized pools (14N containing). Degradation of the old pool of protein does not mean that there will be a loss of total protein. Moreover, immunoblotting cannot distinguish old and newly synthesized pools of protein. Where overall peptide counts are listed for each protein identified at both time points. As peptides derive from proteins, the table provided with the manuscript reflects what immunoblotting would, but on a larger and more precise scale.

Comment 3: I think those proteins that can be identified by MIMS at the time point of 6 months but not 10 months deserve more analyses as they might be the key molecules that drive ovarian aging.

Response 3: This comment conflicts with comment 2 from Reviewer #3 (Recommendations For The Authors). This underscores that different researchers will prioritize the value and follow up of such rich datasets differently. We agree that the LLP identified at 6 months are of particular interest to reproductive aging, and we are planning to follow up on these in future studies.

Comment 4: Figure 1 – figure supplement 1 C-F, compared with the published literature, the numbers of follicles at different developmental stages and ovulated oocytes at both ages of 6 months and 10 months were dramatically low in this study. For 6-month-old female mice, the reproductive aging just begins, thus these numbers should not be expected to decrease too much. In addition, follicle counting was carried out only in an area of a single section, which is an inaccurate way, because the numbers and types of follicles in various sections differ greatly. Also, the data from a single section could not represent the changes in total follicle counts.

Response 4: We have addressed these points in response to Comment 1 in the Reviewer #2 Public Review, and corresponding changes in the text have been noted.

Comment 5: The study lacks follow-up verification experiments to validate their MIMS data.

Response 5: Two independent mass spectrometry approaches (MIMS and LC-MS/MS) were used to validate the presence of long-lived macromolecules in the ovary and oocyte. Studies focused on the role of specific long-lived proteins in oocyte and ovarian biology as well as how they change with age in terms of function, turnover, and modification are beyond the scope of the current study but ongoing. We have acknowledged these important next steps in the manuscript text (lines 286-288 and 311-312).

**Reviewer #3 (Recommendations For The Authors):**
Comment 1: The authors used the 6-month mice group to represent the aged model, and examined the LLPs from 1 month to 6 months. Indeed, 6-month-old mice start to show age-related changes; however, for the reproductive aging model, the most widely accepted model is that 10-month-old age mice start to show reproductive-related changes and 12-month-old mice (corresponding to 35-40 year-old women) exhibit the representative reproductive aging phenotypes. Therefore, the data may not present the typical situation of LLPs during reproductive aging.

Response 1: As described in the response to Comment 1 in the Reviewer #3 Public Review, there were clear logistical and technical feasibility reasons why the 6 month and 10-month timepoints were selected for this study. Importantly, however, these timepoints do represent a reproductive aging continuum as evidenced by age-related changes in multiple parameters. Furthermore, there were ultimately very few LLPs that remained at 10 months in both the oocyte and ovary, so inclusion of the 6-month time point was an important intermediate. Whether the LLPs at the 6-month timepoint serve as a protective mechanism in maintaining gamete quality or whether they contribute to decreased quality associated with reproductive aging is an intriguing dichotomy which will require further investigation. This has been added to the discussion (lines 247-257).

Comment 2: Following the point above, the authors examined the ovaries in 6 months and 10 months mice by proteomics, and found that 6 months LLPs were not identical compared with 10 months, while there were Tubb5, Tubb4a/b, Tubb2a/b, Hist2h2 were both expressed at these two time points (Fig 2B), why the authors did not explore these proteins since they expressed from 1 month to 10 months, which are more interesting.

Response 2: The objective of this study was to profile the long-lived proteome in the ovary and oocyte as a resource for the field rather than delving into specific LLPs at a mechanistic level. That being said, we wholeheartedly agree with the reviewer that the proteins that were identified at both 6 month and 10 months are the most robust and long lived and worthy of prioritizing for further study. Interestingly, Tubb5 and Tubb4a have high homology to primate-specific Tubb8, and Tubb8 mutations in women are associated with meiosis I arrest in oocytes and infertility (Dong et al., 2023; Feng et al., 2016). Thus, perturbation of these specific proteins by virtue of their long-lived nature may be associated with impaired function and poor reproductive outcomes. We have highlighted the importance of these LLPs which are present at both timepoints and persist to at least 10 months in the manuscript text (lines 259-270).

Comment 3: The authors also need to provide a hypothesis or explanation as to why LLDs from 6 months LLPs were not identical compared with 10 months.

Response 3: We agree that LLDs identified at 10 months should be also identified as long-lived at 6 months. This is a common limitation of mass spectrometry-based proteomics where each sample is prepared and run individually, which introduces variability between biological replicates, especially when it comes to low abundant proteins. It is key to note that just because we do not identify a protein, it does not mean the protein is not there – it merely means that we were not able to detect it in this particular experiment, but low levels of the protein may still be there. To compensate for this known and inherent variability, we have applied stringent filtering criteria where we required long-lived peptides to be identified in an independent MS scan (alternative is to identify peptide in either heavy or light scan and use modeling to infer FA value based on m/z shift), which gave us peptides of highest confidence. Ideally, these experiments would be done using TMT (tandem mass tag) approach. However, TMT-based experiments typically require substantial amount of input (80-100ug per sample) which unfortunately is not feasible with oocytes obtained from a limited number of pulse-chased animals. We have added this explanation to the discussion (lines 265-270).

Comment 4: The reviewer thinks that LLPs from 6 months to 10 months may more closely represent the long-lived proteins during reproductive aging.

Response 4: We fully agree that understanding the identity of LLPs between the 6 month and 10 month period will be quite informative given that this is a dynamic period when many of LLPs get degraded and thus might be key to the observed decline in reproductive aging. This is a very important point that we hope to explore in future follow-up studies.

Comment 5: The authors used proteomics for the detection of ovaries and oocytes, however, there are no validation experiments at all. Since proteomics is mainly for screening and prediction, the authors should examine at least some typical proteins to confirm the validity of proteomics. For example, the authors specifically emphasized the finding of ZP3, a protein that is critical for fertilization.

Response 5: Thank you, we agree that closer examination of proteins relevant and critical for fertilization is of importance. However, a detailed analysis of specific proteins fell outside of the scope of this study which aimed at unbiased identification of long-lived macromolecules in ovaries and oocytes. We hope to continue this important work in near future.

Comment 6: For the oocytes, the authors indicated that cytoskeleton, mitochondria-related proteins were the main LLPs, however, previous studies reported the changes of the expression of many cytoskeleton and mitochondria-related proteins during oocyte aging. How do the authors explain this contrary finding?

Response 6: Our findings are not contrary to the studies reporting changes in protein expression levels during oocyte aging – the two concepts are not mutually exclusive. The average FA value at 6-month chase for oocyte proteins is 41.3 %, which means that while 41.3% of long-lived proteins pool persisted for 6 months, the other 58.7% has in fact been renewed. With the exception of few mitochondrial proteins (Cmkt2 and Apt5l), and myosins (Myl2 and Myh7), which had FA values close to 100% (no turnover), most of the LLPs had a portion of protein pools that were indeed turned over. Moreover, we included new data analysis illustrating that we identify comparable number of mitochondrial proteins between the two time points, indicating that while the long-lived pools are changing over time, the total content remains stable (Figure 3 – figure supplement 1E-G).

Comment 7: The authors also should provide in-depth discussion about the findings of the current study for long-lived proteins. In this study, the authors reported the relationship between these "long-lived" proteins with aging, a process with multiple "changes". Do long-lived proteins (which are related to the cytoskeleton and mitochondria) contribute to the aging defects of reproduction? or protect against aging?

Response 7: This is a very important comment and one that needs further exploration. The fact is – we do not know at this moment whether these proteins are protective or deleterious, and such a statement would be speculative at this stage of research into LLPs in ovaries and oocytes. Future work is needed to address this question in detail.

Briley, S. M., Jasti, S., McCracken, J. M., Hornick, J. E., Fegley, B., Pritchard, M. T., & Duncan, F. E. (2016). Reproductive age-associated fibrosis in the stroma of the mammalian ovary. *Reproduction*, *152*(3), 245-260. https://doi.org/10.1530/REP-16-0129

Chiang, T., Duncan, F. E., Schindler, K., Schultz, R. M., & Lampson, M. A. (2010). Evidence that Weakened Centromere Cohesion Is a Leading Cause of Age-Related Aneuploidy in Oocytes. *Current Biology*, *20*(17), 1522-1528. https://doi.org/10.1016/j.cub.2010.06.069

Dong, J., Jin, L., Bao, S., Chen, B., Zeng, Y., Luo, Y., Du, X., Sang, Q., Wu, T., & Wang, L. (2023). Ectopic expression of human TUBB8 leads to increased aneuploidy in mouse oocytes. *Cell Discov*, *9*(1), 105. https://doi.org/10.1038/s41421-023-00599-z

Duncan, F. E., Jasti, S., Paulson, A., Kelsh, J. M., Fegley, B., & Gerton, J. L. (2017). Age-associated dysregulation of protein metabolism in the mammalian oocyte. *Aging Cell*, *16*(6), 1381-1393. https://doi.org/10.1111/acel.12676

Feng, R., Sang, Q., Kuang, Y., Sun, X., Yan, Z., Zhang, S., Shi, J., Tian, G., Luchniak, A., Fukuda, Y., Li, B., Yu, M., Chen, J., Xu, Y., Guo, L., Qu, R., Wang, X., Sun, Z., Liu, M., . . . Wang, L. (2016). Mutations in TUBB8 and Human Oocyte Meiotic Arrest. *N Engl J Med*, *374*(3), 223-232. https://doi.org/10.1056/NEJMoa1510791

Fornasiero, E. F., & Savas, J. N. (2023). Determining and interpreting protein lifetimes in mammalian tissues. *Trends Biochem Sci*, *48*(2), 106-118. https://doi.org/10.1016/j.tibs.2022.08.011

Hark, T. J., & Savas, J. N. (2021). Using stable isotope labeling to advance our understanding of Alzheimer's disease etiology and pathology. *J Neurochem*, *159*(2), 318-329. https://doi.org/10.1111/jnc.15298

Kerr, J. B., Hutt, K. J., Michalak, E. M., Cook, M., Vandenberg, C. J., Liew, S. H., Bouillet, P., Mills, A., Scott, C. L., Findlay, J. K., & Strasser, A. (2012). DNA damage-induced primordial follicle oocyte apoptosis and loss of fertility require TAp63-mediated induction of Puma and Noxa. *Mol Cell*, *48*(3), 343-352. https://doi.org/10.1016/j.molcel.2012.08.017

Kimler, B. F., Briley, S. M., Johnson, B. W., Armstrong, A. G., Jasti, S., & Duncan, F. E. (2018). Radiation-induced ovarian follicle loss occurs without overt stromal changes. *Reproduction*, *155*(6), 553-562. https://doi.org/10.1530/REP-18-0089

Kirkland, J. L. (2013). Translating advances from the basic biology of aging into clinical application. *Exp Gerontol*, *48*(1), 1-5. https://doi.org/10.1016/j.exger.2012.11.014

Mara, J. N., Zhou, L. T., Larmore, M., Johnson, B., Ayiku, R., Amargant, F., Pritchard, M. T., & Duncan, F. E. (2020). Ovulation and ovarian wound healing are impaired with advanced reproductive age. *Aging (Albany NY)*, *12*(10), 9686-9713. https://doi.org/10.18632/aging.103237

Perrone, R., Ashok Kumaar, P. V., Haky, L., Hahn, C., Riley, R., Balough, J., Zaza, G., Soygur, B., Hung, K., Prado, L., Kasler, H. G., Tiwari, R., Matsui, H., Hormazabal, G. V., Heckenbach, I., Scheibye-Knudsen, M., Duncan, F. E., & Verdin, E. (2023). CD38 regulates ovarian function and fecundity via NAD(+) metabolism. *iScience*, *26*(10), 107949. https://doi.org/10.1016/j.isci.2023.107949

Quan, N., Harris, L. R., Halder, R., Trinidad, C. V., Johnson, B. W., Horton, S., Kimler, B. F., Pritchard, M. T., & Duncan, F. E. (2020). Differential sensitivity of inbred mouse strains to ovarian damage in response to low-dose total body irradiationdagger. *Biol Reprod*, *102*(1), 133-144. https://doi.org/10.1093/biolre/ioz164

Savas, J. N., Toyama, B. H., Xu, T., Yates, J. R., 3rd, & Hetzer, M. W. (2012). Extremely long-lived nuclear pore proteins in the rat brain. *Science*, *335*(6071), 942. https://doi.org/10.1126/science.1217421

Toyama, B. H., Savas, J. N., Park, S. K., Harris, M. S., Ingolia, N. T., Yates, J. R., 3rd, & Hetzer, M. W. (2013). Identification of long-lived proteins reveals exceptional stability of essential cellular structures. *Cell*, *154*(5), 971-982. https://doi.org/10.1016/j.cell.2013.07.037